# Association of Muscle Mass, Muscle Strength, and Muscle Function with Gait Ability Assessed Using Inertial Measurement Unit Sensors in Older Women

**DOI:** 10.3390/ijerph19169901

**Published:** 2022-08-11

**Authors:** Bohyun Kim, Changhong Youm, Hwayoung Park, Myeounggon Lee, Hyejin Choi

**Affiliations:** 1Department of Health Sciences, The Graduate School of Dong-A University, Busan 49315, Korea; 2Department of Health Care and Science, Dong-A University, Busan 49315, Korea; 3Interdisciplinary Consortium on Advanced Motion Performance (iCAMP), Michael E. DeBakey Department of Surgery, Baylor College of Medicine, Houston, TX 77030, USA

**Keywords:** fall, gait variability, inertial measurement unit, muscle atrophy, older women

## Abstract

Aging-related muscle atrophy is associated with decreased muscle mass (MM), muscle strength (MS), and muscle function (MF) and may cause motor control, balance, and gait pattern impairments. This study determined associations of three speed-based gait variables with loss of MM, MS, and MF in older women. Overall, 432 older women aged ≥65 performed appendicular skeletal muscle, handgrip strength, and five times sit-to-stand test to evaluate MM, MS, and MF. A gait test was performed at three speeds by modifying the preferred walking speed (PWS; slower walking speed (SWS); faster-walking speed (FWS)) on a straight 19 m walkway. Stride length (SL) at PWS was significantly associated with MM. FWS and coefficient of variance (CV) of double support phase (DSP) and DSP at PWS showed significant associations with MS. CV of step time and stride time at SWS, FWS, and single support phase (SSP) at PWS showed significant associations with MF. SL at PWS, DSP at FWS, CV of DSP at PWS, stride time at SWS, and CV of SSP at PWS showed significant associations with composite MM, MS, and MF variables. Our study indicated that gait tasks under continuous and various speed conditions are useful for evaluating MM, MS, and MF.

## 1. Introduction

Aging-related muscle atrophy is associated with decreased number and size of muscle fibers due to gradual motor neuron loss, which reduces muscle mass (MM) and strength (MS) [1]. Furthermore, motor neuron loss gradually reduces muscle function (MF) [1], resulting in motor control, balance, and gait pattern impairments [2], leading to an increased risk of falls [3]. Sarcopenia and frailty, characterized by decreased MM, MS, and MF, are associated with decreased gait ability [4,5,6]. Gait analysis effectively identifies early pathology, assesses disease progression, and predicts fall risk [7,8]. Therefore, gait analysis may help predict and identify decreased MM, MS, and MF [4,5,6].

Previous studies on gait ability with decreasing MM reported decreased gait speed and increased step-time variability [2]. Studies on gait ability with decreasing MS reported the following: decreased gait speed, stride length, and swing time; increased stance time, stride time, double support time; variability of stride length, swing time, double support time [9], and step width [10]. However, previous studies have repeatedly measured short walkways of 3–10 m [2,10], and it is questionable whether measurements obtained by these methods are similar to the actual gait pattern [11]. Recently, continuous gait analysis of at least 30–40 steps was suggested to increase gait measurement reliability [11,12]. Therefore, analysis of various spatiotemporal gait parameters of continuous walking steps may improve risk prediction and classification sensitivity and specificity according to MM, MS, and MF loss [4,11,12].

Recent studies using inertial measurement unit (IMU) sensors were conducted as an alternative for gait analysis, and their validity and reliability were determined in patients and healthy individuals [13,14]. Furthermore, an analysis of gait characteristics at various speeds may be more sensitive to age-related decline by placing greater demand on motor control [15,16]. As reported, slower and faster speeds are significantly related to advanced age in older adults [15,16] and MS loss [9]. However, few studies have comprehensively examined the association of MM, MS, and MF with gait ability at continuous and varying speeds. Therefore, this study aimed to analyze the association between three speed-based gait variables (using an IMU sensor) and MM, MS, and MF loss in older women. We hypothesized that reduced MM, MS, and MF would worsen gait stability with slower and shortened patterns and worsen gait variability (GV) under three-speed conditions. Moreover, GV variables were assumed to be significantly related to MM, MS, and MF.

## 2. Materials and Methods

### 2.1. Participants

Participants were recruited through a community-wide survey conducted in Busan Metropolitan City. Of the 600 community-dwelling women aged ≥65 years contacted, 530 responded (response rate: 88.3%). We included participants who could walk on their own without any support. Participants with histories of musculoskeletal injuries or neurophysiological problems and cardiovascular or pulmonary diseases that could affect gait and physical ability in the past six months were excluded. Forty-nine participants were excluded according to the age range and inclusion and exclusion criteria. Fifty-one participants were excluded from the study for the following reasons: 12 did not participate in the test for personal reasons, 18 declined to participate, and 21 did not complete trials. Finally, 430 older women aged 65–89 years participated in the study (Figure 1). All participants provided written informed consent. The institutional review board of Dong-A University approved this study (IRB number: 2–104709–AB–N–01–201808–HR–023–02).

### 2.2. Assessment of MM, MS, and MF

MM was assessed using bioelectrical impedance analysis (InBody 270, Biospace, Seoul, Korea). Appendicular skeletal muscle (ASM) was quantified by summing the lean mass of both arms and legs. MM was defined by dividing ASM by the square of the height (ASM/height^2^). MS was measured using handgrip strength and an isometric digital handgrip dynamometer (TKK 5401 Grip-D, Takei Scientific Instruments Co., Ltd., Tokyo, Japan). The maximum of two dominant arm measurements was recorded. MF was performed by five times sit-to-stand (STS) test to identify MS and power (older adults’ two primary lower limb functional abilities) [17], and the time required was recorded. All assessments were based on the sarcopenia diagnostic criteria of the Asian Working Group [18]. The integrated variables of MM, MS, and MF were defined as summation after Z-normalization to determine overall muscle condition. MF was calculated after reversing the value because an increase in measurement time indicates decreased function.

### 2.3. Assessment of Gait Performance

All participants performed three gait performance tests by modifying their preferred walking speed (PWS; slower walking speed (SWS), 80% of PWS; faster-walking speed (FWS), 120% of PWS) on a straight 19 m overground walkway [19]. PWS was defined using a metronome (beats/min). Participants practiced walking for approximately 10 min under three-speed conditions using a metronome in a familiarization session. They were asked to walk as close as possible to the metronome’s targeted SWS and FWS. We excluded the 2 m sections of acceleration and deceleration step periods of the gait performance test to analyze the steady-state condition (Figure 2).

### 2.4. Instrumentation

Gait analysis was evaluated using a gait analysis system (DynaStab, JEIOS, Busan, Korea), including shoe-type data loggers (Smart Balance SB-1, JEIOS, Busan, Korea) with embedded IMU (IMU-3000, InvenSense, San Jose, CA, USA) on both outsoles. Gait data were measured by triaxial accelerations up to ±6 g and triaxial angular velocities up to ±500°s^−1^ along three orthogonal axes. Measured data were transmitted to the gait analysis system using Bluetooth wireless connection and collected at a 100 Hz sampling frequency utilizing a data acquisition system (Smart Balance version 1.5, JEIOS, Busan, Korea) [13].

### 2.5. Data Analysis and Statistical Analysis

Gait data were filtered to a cutoff frequency of 10 Hz using a second-order Butterworth low-pass filter [13]. Heel-strike and toe-off of gait events were detected when linear acceleration along anteroposterior and vertical axes reached their maximum [13]. Next, we calculated spatiotemporal parameters, including (1) pace: walking speed, stride length, and step length; (2) rhythm: total steps, stride time, and step time; (3) phases: single support, double support, and stance. For GV, spatiotemporal parameters were quantified as the coefficient of variance (CV; standard deviation/mean × 100) [20].

All statistical analyses were conducted using IBM SPSS Statistics for Windows, version 21.0 (IBM Corp., Armonk, NY, USA). The Shapiro–Wilk test was used to examine the normal distribution of data. Intraclass correlation coefficient (ICC) analysis (2,1) was conducted to confirm reliability between relatively calculated and executed slow and fast speed values through the participant’s preferred speed. Limits of agreement (LoAs) between measured and estimated gait speeds were calculated using the Bland–Altman plot [21]. Before additional analysis, all variables were Z-normalized (value–mean/standard deviation). Pearson’s product-moment correlation analysis was performed to determine the correlation of dependent variables (MM, MS, MF, and composite MM, MS, and MF) with gait variables. Furthermore, stepwise multivariable linear regression analysis was performed to identify independent factors explaining dependent variables. Covariates included age, height, and body mass.

All dependent variables were categorized as quartiles, and the first and fourth quartiles were defined as low and high groups, respectively. Binary logistic regression analysis was performed using the forward variable selection method to determine the classifiers between high and low groups. Moreover, to investigate the classification accuracy of low and high groups, areas under the curve (AUCs) were calculated using receiver operating characteristic curve analysis, and sensitivity, specificity, and cutoff values of variables that could distinguish between low and high groups were analyzed. Statistical significance was set at *p* < 0.05.

## 3. Results

### 3.1. Participant Demographic, MM-, MS-, and MF-Related, and Gait Variables

Participants’ demographic and MM-, MS-, and MF-related characteristics and reliability of SWS and FWS are shown in Table 1. As shown in Figure 3, the participants’ degree of agreement at slower and faster speeds was 94.4% and 95.1%, respectively.

### 3.2. Relationship between Gait with MM-, MS-, and MF-Related Variables

MM showed a positive correlation with walking speed, stride length, and step length at PWS and FWS. MM showed a negative correlation with CVs of stride length, stride time, and stance phase at FWS, and CVs of step length, step time, single support phase, and stance phase at PWS and FWS.

MS showed a positive correlation with walking speed, stride and step lengths, and single support phase under three-speed conditions. Conversely, MS showed a negative correlation as follows: total steps, double support phase, stance phase, CVs of stride length and time, and single support phase under three-speed conditions; stride and step times at PWS and FWS; CV of step length and time at SWS and PWS; CV of double support phase at PWS; CV of stance phase at FWS.

MF showed a positive correlation with walking speed, stride and step lengths, and single support phase under three-speed conditions and stride and step times at SWS. MF was negatively correlated as follows: total steps, double support phase, CVs of stride and step lengths, stride and step times, and stance phase under three-speed conditions; stance phase and CV of single support phase at PWS and FWS; CV of double support phase at SWS; stride time at FWS.

The integrated variables of MM, MS, and MF were positively correlated with walking speed, stride length, and step length under three-speed conditions, single support phase at PWS and FWS, and stride and step times at SWS. Composite MM, MS, and MF variables showed a negative correlation as follows: total steps and CVs of stride and step lengths, stride and step times, and stance phase under three-speed conditions; stride time, double support phase, stance phase, and CV of single support phase at PWS and FWS; step time at FWS; CV of double support phase at PWS (Figure 4).

### 3.3. Association of Gait with MM-, MS-, and MF-Related Variables

Table 2 lists only statistically significant results for the association of gait variables at three different speeds with MM-, MS-, and MF-related variables in older women. After adjusting for confounders, stride length at PWS (β = 0.098, *p* = 0.007) was significantly associated with MM. Walking speed at FWS (β = 0.112, *p* = 0.048), CV of double support phase at PWS (β = −0.120, *p* = 0.004), and double support phase at PWS (β = −0.136, *p* = 0.015) were significantly associated with MS. CV of step time at SWS (β = −0.183, *p* < 0.001), stride time at SWS (β = 0.258, *p* < 0.001), walking speed at FWS (β = 0.311, *p* < 0.001), and CV of single support phase at PWS (β= −0.084, *p* = 0.034) were significantly associated with MF. Stride length at PWS (β = 0.182, *p* = 0.003), double support phase at FWS (β = −0.203, *p* = 0.001), CV of double support phase at PWS (β = −0.122, *p* = 0.009), stride time at SWS (β = 0.117, *p* = 0.019), and CV of single support phase at PWS (β = −0.105, *p* = 0.038) were significantly associated with the integrated variables of MM, MS, and MF.

### 3.4. Classifier Variables for the Low and High Groups

Stepwise binary logistic regression analysis of high and low MM groups revealed that stride length at PWS differed significantly. For MS, double support phase in PWS (cutoff value: 15.16%; AUC: 0.681; *p* < 0.001; sensitivity: 0.633; specificity: 0.635) was significantly different between the groups. For MF, walking speed at FWS and stride time at SWS differed significantly between groups. For the integrated variables of MM, MS, and MF, stride length at PWS, CV of single support phase at PWS (cutoff value: 2.29%; AUC: 0.737, *p* < 0.001; sensitivity: 0.676; specificity: 0.686), and CV of double support phase at PWS (cutoff value: 9.83%; AUC: 0.589, *p* = 0.023; sensitivity: 0.552; specificity: 0.562) differed significantly between groups (Table 3, Figure 5).

## 4. Discussion

The main findings of this study are as follows: (1) stride length at PWS was associated with MM; (2) walking speed at FWS and double support phase and CV of double support phase at PWS were associated with MS; (3) walking speed at FWS, stride time and CV of step time at SWS, and CV of single support phase at PWS were associated with MF; (4) stride length and CVs of single and double support phases at PWS, stride time at SWS, and double support phase at FWS were associated with the integrated variables of MM, MS, and MF; (5) stride length and CVs of single and double support phases at PWS were significant in distinguishing between low and high groups of the integrated variables of MM, MS, and MF.

Older postmenopausal women are deficient in sex hormones such as estrogen, which may cause reduced MM and MS [22]. Low MM can generally weaken the muscles and may increase the risk of functional decline [23], while low MS can induce changes in the lower limb joint kinetics and kinematics [24]. Our results showed a strong dependence on MM for stride length at PWS. Previous studies have described changes in stride parameters in older people as an increase in energy cost during walking [25], compensation for muscle weakness [26], and mobility impairment [27]. Furthermore, weakened hip extensor and ankle plantar flexor muscles could reduce the body’s forward ability during gait initiation, thereby reducing the overall stride length of the gait cycle [9,28]. Therefore, we suggest that stride length at PWS may predict a decrease in MM in older women.

Furthermore, our results indicated that MS was positively associated with walking speed at FWS and negatively associated with the double support phase and CV of the double support phase at PWS. FWS has previously been reported as a walking condition that can optimize the detection of challenging and high-level walking disorders in older people [15,16]. It requires high muscle activity for propulsion and stability to increase joint range of motion for longer steps and higher cadence and to increase demand for the function of eccentric muscles and for shock absorption [29,30]. Therefore, an older individual with reduced MS may not generate appropriate ankle power during push-off at FWS [24]. Additionally, MS loss affects posture and movement by lowering the mediolateral momentum of the center of mass by increasing the double support phase and step width to maintain a stable walking pattern [24]. Callisaya et al. [31] reported that double support time variability is related to dynamic balance during gait and depends on proprioceptive feedback to maintain a consistent timing during the double support phase. Therefore, changes in gait patterns with MS loss in older people are related to dynamic balance during walking and may increase the risk of falling.

The STS task evaluates the MF of the lower limbs [18] and is related to a decrease in gait speed [32]. Our study revealed that MF was positively associated with walking speed at FWS and stride time at SWS and negatively associated with the CV of step time at SWS and CV of single support phase at PWS. SWS requires a strategy to increase the mediolateral displacement of the center of mass to maintain dynamic balance and increase the support base [33]. Previous studies have reported that SWS is an attention-demanding task owing to reduced gait automaticity, higher cortical control, and changes in muscle activity patterns [15,33,34]. Our results indicate that despite the increase in MF, the stride time increases in SWS, the decrease in MF and the increase in CV of step time in SWS are associated. Therefore, despite their high MF, SWS may be a challenging task for controlling motor function in older women.

Additionally, we attempted to determine the association of composite MM, MS, and MF variables with gait variables. The integrated variables of MM, MS, and MF were positively associated with stride length at PWS and stride time at SWS and negatively associated with double support phase at FWS and CVs of double and single support phase at PWS. These results were similar to the gait pattern changes in MM, MS, and MF variables. Interestingly, stepwise binary logistic regression analysis for high and low groups the integrated variables of MM, MS, and MF revealed significant differences in stride length and CVs of single and double support phases at PWS. GV is defined as stride-to-stride fluctuations in gait [35] and is related to efficient gait control and safety [36]. Previous studies have reported the relationship between low MM, MS, and MF with increased GV [2,10,37]. Increased GV may be caused by the loss of lower limb strength and range of motion, increased change in muscle activation, and decreased balance [37,38]. This difference may be due to the lack of postural stability associated with less stable force output during walking, where reduced dynamic walking stability relies on motor control and reduction in automatic stepping mechanisms [39]. Montero-Odasso et al. [40] suggested that subtle impairments may cause higher GV values in frail populations with higher cerebral functions and cognition. High GV requires a high level of motor cortical control and attention and may be due to an impairment of the basal ganglia and central nervous system function with aging [39]. Therefore, we suggest that stride length and CVs of single and double support phases at PWS may predict decreased composite MM, MS, and MF variable in older women.

Our study confirmed an association of MM, MS, and MF with gait variables in older women. In addition, an association of the integrated variables of MM, MS, and MF with decreased gait ability was also confirmed. However, this study has several limitations. First, using specific questionnaires, our study did not consider the effects of important older adult characteristics, such as frailty status, drug dosage, and fear of falling. Second, all participants performed the walking test at a controlled speed. However, we confirmed the ICC results, which were calculated to distinguish between measured and estimated speed consistencies at SWS and FWS. In addition, we collected continuous walking step data using a 19 m walkway to increase measurement reliability.

## 5. Conclusions

Our study indicated that a decrease in MM, MS, and MF is associated with a decrease in gait ability. Stride length at PWS may predict a reduction in MM in older women. Walking speed at FWS and double support phase and CV of double support phase at PWS were associated with MS. Walking speed at FWS, stride time and CV of step time at SWS, and CV of single support phase at PWS were associated with MF. Hence, low MM, MS, and MF were associated with a decline in gait ability based on the three speeds. Finally, stride length and CVs of single and double support phases at PWS, stride time at SWS, and double support phase at FWS were associated with composite MM, MS, and MF variable. In particular, stride length and CVs of single and double support phases at PWS can distinguish between low and high groups of composite MM, MS, and MF variable. Therefore, we suggest that gait tasks under continuous and varying speed conditions are useful for evaluating MM, MS, and MF, which are important for the daily life activities of older women. Older women who are vulnerable to decreased MM, MS, and MF may need intervention programs to prevent falls and improve their gait ability and motor function.

## Figures and Tables

**Figure 1 ijerph-19-09901-f001:**
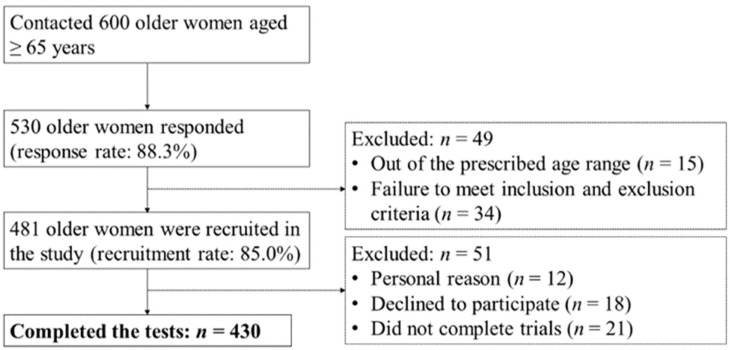
Flow diagram of the participant recruitment process.

**Figure 2 ijerph-19-09901-f002:**
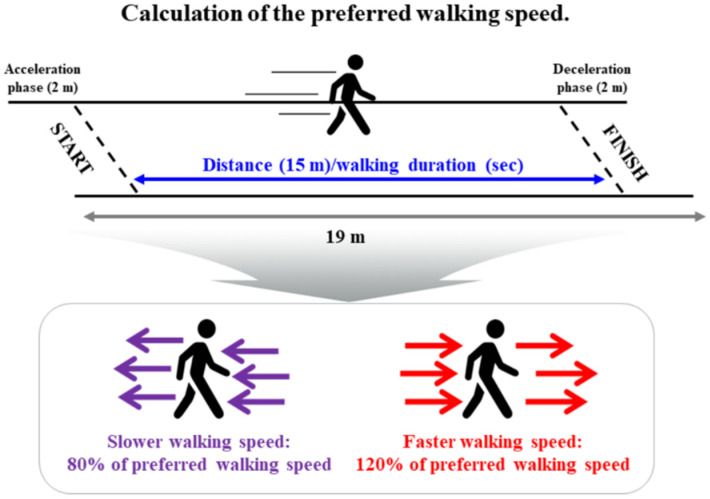
Chart showing the gait test: three-speed conditions based on the preferred walking speed (slower walking speed (80% of preferred walking speed) and faster walking speed (120% of preferred walking speed)).

**Figure 3 ijerph-19-09901-f003:**
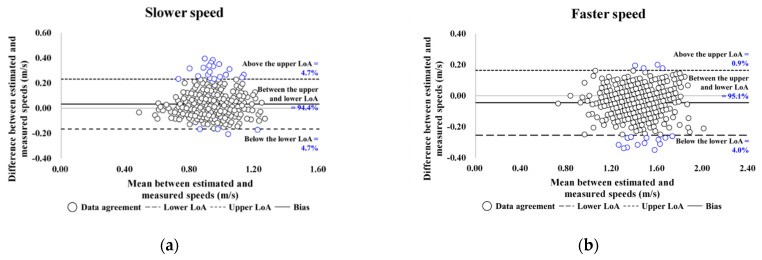
Bland–Altman plots for data agreement between the estimated and measured overground walking speeds: (**a**,**b**) are the slower and faster speed results for older women. LoA, limits of agreement.

**Figure 4 ijerph-19-09901-f004:**
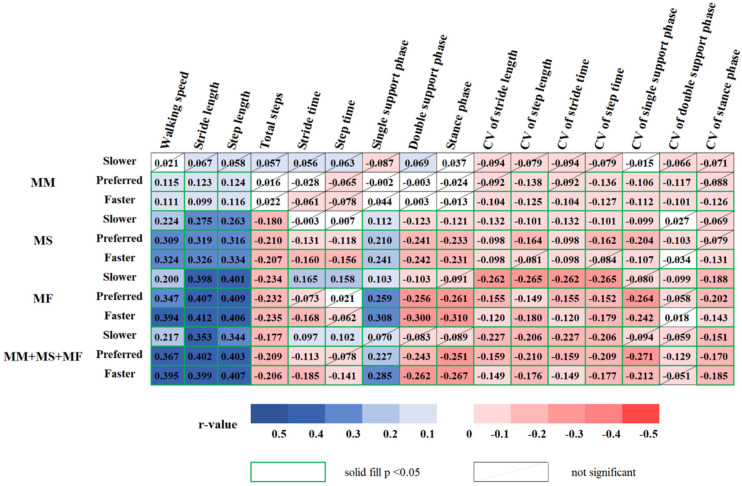
Correlogram representing the relationships between the MM-, MS-, and MF-related variables and three speed-based gait variables in older women: blue represents positive correlation, and red represents negative correlation, while the non-significant results (*p* < 0.05) are crossed out. MM: muscle mass; MS: muscle strength; MF: muscle function; CV: coefficient of variance.

**Figure 5 ijerph-19-09901-f005:**
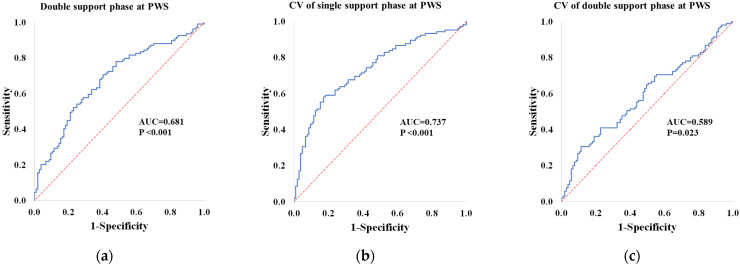
Receiver operating characteristic (ROC) curves of the double support phase and CVs of single and double support phases at PWS: The AUC and *p*-values of the ROC curves are written in bold style at the bottom-right corner of each panel. AUC: area under the curve; PWS: preferred walking speed; CV: coefficient of variance.

**Table 1 ijerph-19-09901-t001:** Participants’ demographic and muscle-related characteristics (*n* = 430).

Variables	All Participants
Age (years)	72.39 ± 4.94
Height (cm)	153.06 ± 5.30
Body mass (kg)	58.21 ± 8.34
BMI (kg/m^2^)	24.95 ± 3.12
BFP (%)	34.47 ± 5.69
Skeletal muscle mass (kg)	20.41 ± 2.55
MM (kg/m^2^)	8.69 ± 0.80
MS (kg)	22.02 ± 4.14
MF (s)	9.81 ± 3.47
SWS	
Estimated/measured (m/s)	0.94/0.90
ICC (2,1)	0.825
*p*-value	<0.001
FWS	
Estimated/measured (m/s)	1.41/1.45
ICC (2,1)	1.000
*p*-value	<0.001

Data are presented as mean ± standard deviation. BMI, body mass index; BFP, body fat percentage; MM, muscle mass; MS, muscle strength; MF, muscle function; SWS, slower walking speed; FWS, faster walking speed; ICC, intraclass correlation coefficient.

**Table 2 ijerph-19-09901-t002:** Association of the gait variables at three different speeds with MM, MS, and MF-related variables.

Variables	β (SE)	T	*p*-Value
**MM**	**R^2^ = 0.345**		
Stride length (preferred)	0.098 (0.036)	2.732	0.007
**MS**	**R^2^ = 0.265**		
Walking speed (faster)	0.112 (0.057)	1.984	0.048
CV of double support phase (preferred)	−0.120 (0.042)	–2.879	0.004
Double support phase (preferred)	−0.136 (0.056)	−2.445	0.015
**MF**	**R^2^ = 0.300**		
CV of step time (slower)	−0.183 (0.038)	−4.805	<0.001
Stride time (slower)	0.258 (0.040)	6.476	<0.001
Walking speed (faster)	0.311 (0.044)	7.118	<0.001
CV of single support phase (preferred)	−0.084 (0.040)	−2.122	0.034
**MM + MS + MF**	**R^2^ = 0.330**		
Stride length (preferred)	0.182 (0.061)	2.973	0.003
Double support phase (faster)	−0.203 (0.059)	−3.475	0.001
CV of double support phase (preferred)	−0.122 (0.046)	−2.636	0.009
Stride time (slower)	0.117 (0.050)	2.354	0.019
CV of single support phase (preferred)	−0.105 (0.050)	−2.081	0.038

The model was adjusted for age, height, and body mass. MM, muscle mass; MS, muscle strength; MF, muscle function; CV, coefficient of variance; SE, standard error; significant difference, *p* < 0.05.

**Table 3 ijerph-19-09901-t003:** Binary logistic regression results for the high and low groups according to the quartiles.

Variables		β (SE)	OR (95% CI)	*p*-Value	R_N_^2^
MM	Stride length (preferred)	0.639 (0.284)	1.895 (1.086–3.306)	0.024	0.705
MS	Double support phase (preferred)	−0.563 (0.192)	0.569 (0.391–0.830)	0.003	0.426
MF	Walking speed (faster)	1.145 (0.222)	3.144 (2.036–4.856)	<0.001	0.402
	Stride time (slower)	0.783 (0.204)	2.187 (1.466–3.263)	<0.001	
MM + MS + MF	Stride length (preferred)	1.063 (0.283)	2.896 (1.664–5.040)	<0.001	0.591
	CV of single support phase (preferred)	−0.514 (0.234)	0.598 (0.378–0.947)	0.028	
	CV of double support phase (preferred)	−0.428 (0.202)	0.652 (0.438–0.968)	0.034	

The model was adjusted for age, height, and body mass. MM, muscle mass; MS, muscle strength; MF, muscle function; CV, coefficient of variance; SE, standard error; OR, odds ratio; CI, confidence interval; R_N_^2^, Model fit statistic Nagelkerke’s R^2^; significant difference, *p* < 0.05.

## Data Availability

The datasets that support the findings of this study are available from the corresponding author upon reasonable request.

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
