# Peer review of "Association of Muscle Mass, Muscle Strength, and Muscle Function with Gait Ability Assessed Using Inertial Measurement Unit Sensors in Older Women"

_ijerph, 2022, doi:10.3390/ijerph19169901_

Round 1

Reviewer 1 Report

In my opinion, you need to talk more about the inverse correlation results of your study. In the discussion when discussing the correlation, indicate whether it is positive or negative. It would be positive to include a second paragraph in the discussion where you discuss your own results, the positive and negative correlations obtained and your appreciation of the results with respect to the gait analysis (like line 258).

Reviewer 2 Report

 A strict correlation between aging-related muscle atrophy ( a term "sarcopenia" is more commonly used  for elderly persons) with gait pattern  impairments is rather evident and widely documented. Therefore, the study results could be predicted easily. However, the present study can be considered as the verification performed on the specific regional women population. The experiments are well designed and the statistical methods properly applied.

Thus, generally taken, in my opinion the paper is interesting and worth publishing.   

Two minor remarks:

Abstract - Methods of MM, MS, and MF assessment should be mentioned.

Exclusion criteria - no cardiovascular or pulmonary diseases that could affect the physical capacity (and gait speed) of the examined women were  taken into consideration.

Round 2

Reviewer 1 Report

Authors made the suggestions changes